# Chinese Cabbage *BrCAP* Has Potential Resistance against *Plasmodiophora brassicae*

## Jiawei Zou, Shiqi Gao, Bo Zhang, Wenjie Ge, Jing Zhang and Ruiqin Ji *

Liaoning Key Laboratory of Genetics and Breeding for Cruciferous Vegetable Crops, College of Horticulture, Shenyang Agricultural University, Shenyang 110866, China; zoujiawei0525@163.com (J.Z.)
* Correspondence: ruiqinji@syau.edu.cn

**Abstract:** Clubroot disease caused by *Plasmodiophora brassicae* Woronin (*P. brassicae*) has seriously influenced the production of *Brassica rapa* crops, but the interaction mechanism between *P. brassicae* and *Brassica rapa* is not clear. In our previous study, a differentially expressed protein, G15, was found between the roots of Chinese cabbage inoculated and un-inoculated with *P. brassicae* through two-dimensional electrophoresis (2-DE) and mass spectrometry, and G15 was matched with Bra011464. In this study, Bra011464 was found to have a 94% percent identity with *Arabidopsis thaliana* CAP, named *BrCAP*. The expression of *BrCAP* was the highest in the root compared with the stems and leaves of Chinese cabbage, and its expression in the roots of Chinese cabbage inoculated with *P. brassicae* was significantly higher than that in the control. The results were verified by real-time quantitative polymerase chain reaction (RT-qPCR) and in situ hybridization. Subcellular localization showed that *BrCAP* was localized on chloroplasts of leaf epidermal cells. To verify the function of *BrCAP*, it was found that the *Arabidopsis thaliana* mutant *cap* was more susceptible to infection with *P. brassicae* than the wild type (WT), which suggested that *BrCAP* has a potential role in the resistance progress of Chinese cabbage to *P. brassicae*.

**Keywords:** *Brassica rapa* L. ssp. *pekinensis*; CAP; clubroot disease; plant-pathogen interaction

## 1. Introduction

Chinese cabbage (*Brassica rapa* L. ssp. *pekinensis*) is a subspecies of *B. rapa* and is considered an economically important cruciferous vegetable in Asia, particularly in China, Korea, and Japan [1]. Clubroot is a serious soil-borne disease of cruciferous crops caused by the biotrophic pathogen *Plasmodiophora brassicae* Woronin [2,3]. *P. brassicae* spores are mononuclear and their outer wall is composed of chitin. *P. brassicae* can host exclusively on cruciferous plants. The infection can be carried out at any growth period of cruciferous plants. In the early stage of infection, there is no obvious difference in the above-ground parts of host plants. In the late stage of infection, the leaf edges of plants gradually turn yellow and wither, until the whole plant dies. When the whole plant dies, dormant spores are formed in the cells of the diseased part, and the spores of *P. brassicae* in the tissues can be released into the soil, seriously harming the next crop [4]. Dormant spores of *P. brassicae* have strong survivability in soil. Studies have shown that the dormant spores of *P. brassicae* in soil can still be induced by the external environment and are still capable of causing disease after induction [5]. *P. brassicae* is highly infectious. A low concentration of *P. brassicae* spores can cause disease, with high transmission speed and multiple modes of transmission, as well as insufficient understanding of the pathogenesis of clubroot disease, making clubroot disease a devastating soil-borne disease worldwide [6]. At present, there are no effective disease resistance measures for clubroot disease, so it is necessary to start from the excavation of disease resistance genes. Many scholars have found some disease-resistant genes by transcriptome [7,8], proteome [9,10], molecular markers [11,12], and other methods, but the role of these genes in the disease resistance mechanism is not clear enough.

Plants defend themselves against biotic stresses through a variety of local, constitutive, and inducible mechanisms [13]. These defense responses are stimulated by the systemic acquired resistance (SAR), induced systemic resistance, and so on [14,15]. Salicylic acid (SA) is a key factor in resistance against specific pathogens [16]. When a plant is infected by a pathogen, the SA content can increase, and the expression of genes encoding PR proteins can be activated by the SA signal pathway [17]. Cysteine-rich secreting protein, Antigen 5, and disease-related Protein 1 (CAP), were identified to play a role in the regulation of host immune attack and infection [18]. Chen and colleagues found that the *CAP* gene (named *PROAtCAPE1*) in *Arabidopsis thaliana* was a salt-responsive gene that was reduced in expression during salt treatment [19]. In addition, many studies have shown that the *CAP* gene was highly expressed in roots of *Arabidopsis thaliana* [20], and that its expression could be induced by various external stresses, such as drought stress [21,22], injury induction [19], salt stress induction [21,23], arsenic treatment [24], and iron deficiency [25]. PR-1 protein was a member of a broader family of proteins, named the CAP superfamily. Disease-course related proteins were first identified in TMV-infected tobacco leaves in 1970. At least 17 families have been identified, PR-1 to PR-17 [26], and PR-1 was one of the most produced proteins in the defense response [27]. The significance of PR-1 protein in plant-microbial interactions is now recognized, and an increasing number of identified pathogen-effector proteins interact directly with PR-1 during infection [28,29]. It can be seen that the regulation mechanism of CAP in the defense signal network is complex. At present, the relationship between *CAP* and clubroot disease is not clear.

In our previous study, a differentially expressed protein, G15, was found between the roots of Chinese cabbage inoculated and un-inoculated with *P. brassicae* through two-dimensional electrophoresis (2-DE) and mass spectrometry. These techniques found that G15 can be matched with Bra011464 [30]. In this study, Bra011464 was found to have a 94% percent identity with *Arabidopsis thaliana* CAP, named *BrCAP*. This research aims to study the role of *BrCAP* in the infection process of *P. brassicae*, and provide clues for *Brassica* crops to resist *P. brassicae* infection.

## 2. Materials and Methods

### 2.1. Plant Materials

The plant material used in the experiment was Chinese cabbage 'SN742'. *A. thaliana* mutant *cap* was purchased from the *Arabidopsis thaliana* Information Resource website (TAIR), and tobacco (*Nicotiana benthamiana*) was provided by Shenyang Agricultural University. The *P. brassicae* used in all the experiments was the physiological race No. 4 and was provided by the vegetable genetics breeding laboratory of Shenyang Agricultural University.

### 2.2. Preparation of P. brassicae Suspension

An amount of 20 g of freshly clubbed roots, collected from harvested Chinese cabbage 'SN742', was cut into small pieces. An amount of 100 mL of sterile water was added and homogenized in a blender (JYL-C022E, Joyoung, China) and then filtered through 8 layers of cheesecloth into the bottle. The concentration of spores was adjusted to $1 \times 10^7$ /mL using a hemacytometer under an ordinary light microscope (Nikon Eclipse 80i, Tokyo, Japan). Finally, the suspension was stored at 4 °C for standby.

### 2.3. Plant Materials Cultivation Method

The 'SN742' seeds of Chinese cabbage were washed with 70% ethanol and sterile water for 1 min each, and then evenly spread in the sterilized Petri dish lined with wet filter paper, placed in the constant temperature incubator at 25 °C in the dark environment for 24 h to promote germination. The germinated seeds were transferred to a MLR-350H incubator (SANYO, Osaka, Japan) to be cultivated for 2–3 days under the conditions of light 16 h/dark 8 h, 25 °C, 60% humidity. When the seeding grew two true leaves, treatment group plants were cultured with Hongland solution containing $1 \times 10^7$/mL suspension of *P. brassicae*, and the control group plants were cultured with Hongland solution without

*P. brassicae*. The culture environment of the two groups was consistent. The growth of the plants was observed under the ordinary light microscope (Nikon Eclipse 80i, Tokyo, Japan) daily, until the root hair was found to be infected by *P. brassicae* spores (around the 14th after inoculation with *P. brassicae*). Then, the seedlings were transferred into a medium composed of soil and matrix (1:1, vol/vol) for cultivation, respectively. About 40 days after inoculation, clubbed roots could be found in the roots of Chinese cabbage. It can be used for subsequent experiments. The infected and uninfected roots caused by *P. brassicae* at different periods after inoculation were sampled for further experimentation.

Thirty seeds, each of wild type Columbia (WT), and the *cap* mutant of *A. thaliana*, were placed in a refrigerator for 3–5 days and vernalized at 4 °C. They were transferred into a sterilized medium composed of peat, perlite, and vermiculite (3:2:1, vol/vol/vol). They were covered with a transparent plastic film and placed at 25 °C; when the seeds germinated and grew 1–2 young leaves, the film was removed. When the seedlings grew about 8 true leaves after 20 days, they were watered with 1/2 MS nutrient solution every other week. The water and nutrient solution should be recycled until the flowering of *A. thaliana*.

### 2.4. Cloning of BrCAP and Bioinformatics Analysis

In our previous study, a candidate protein BrCAP was obtained by two-dimensional electrophoresis (2-DE) and mass spectrometry [29]. Its cDNA sequence was obtained from the Chinese cabbage database according to the obtained amino acid sequence. The full-length cDNA of *BrCAP* was cloned using the primers (P1 in Table S1), designed based on the most similar sequence of *BrCAP* (XM_009121709.2) in the *Brassica* database (www.brassicadb.cn/#/ (accessed on 23 March 2021)). The full length of the *BrCAP* was amplified from the roots' cDNA of Chinese cabbage, and the product was connected to the PGEM-T-Easy vector by T4 ligase. The connected production was sent to Sangon Biotech. (Sangon Biotech, Shanghai, China) for sequencing.

The TMHMM website (http://www.cbs.dtu.dk/services/TMHMM/ (accessed on 4 May 2021)) was used to predict transmembrane domain structure, and NCBI (https://www.Ncbi.nlm.nih.gov/structure/CDD/WRPSB.Cgi (accessed on 4 May 2021)) was used to predict the function structure domain.

### 2.5. Real-Time Quantitative Polymerase Chain Reaction (RT-qPCR)

RNA was extracted from the roots, stems, and leaves of Chinese cabbage 'SN742' after inoculation with *Plasmodiophora brassicae* Woronin (*P. brassicae*) on the 14th and 40th days using pure Total RNA Kit (Tiangen, Beijing, China) according to the manufacturer's instructions. Additionally, un-inoculated materials during the same periods were used as the control. The quality of extracted RNA was detected by 1% agarose gel electrophoresis and reversed into cDNA (Vazyme, Nanjing, China). The expression of *BrCAP* was analyzed by RT-qPCR using QuantStudio6 Real-Time PCR (Thermo Fisher, Waltham, MA, USA), with the kit of UltraSYBR Mixture (CWBIO, Jiangsu, China) and P1/P2 primers (in Table S1). Three replicates were performed for different treatments, and 3 seedlings per experimental unit. Data were calculated by using the $2^{-\Delta\Delta ct}$ method, and the significance analysis was performed by Student's test (* $p \leq 0.05$) or Duncan's multivariate interval test in SPSS 11.5 (IBM, Armonk, NY, USA). Origin Pro 7.5 (Origin Lab Corp., Northampton, MA, USA) was used to produce the graphics.

### 2.6. In Situ Hybridization

On the 14th day and 40th day after inoculation with *P. brassicae*, the roots inoculated with *P. brassicae* and control roots were taken for an in situ hybridization test. The samples were fixed with 4% FAA fixing solution, and then dehydrated in gradient ethanol (70%, 80%, 90%, 95% and 100%), cleared in ethanol: xylene mixtures (3:1, 1:1 and 1:3 [*v/v*]) and embedded in 100% paraffin. A *BrCAP*-specific probe fragment was amplified using primer P4 containing *Pst*I and *Kpn*I enzyme restriction sites (Table S1). Five replicates

were performed for each different treatment (five slices were used for each treatment, and each slice had three to five tissues). The vector pSPT18 were digested with *Pst*I and *Kpn*I, purified and connected with a *BrCAP*-specific probe fragment to obtain pSPT18-*BrCAP* recombinant vector. pSPT18-*BrCAP* was purified and digested with *Pst*I or *Kpn*I, and labeled with digoxigenin (DIG) using a SP6/T7 Transcription Kit (Roche, Basle, Switzerland) to synthesize the sense or antisense probes. Sample sections were hybridized with specific DIG-labeled RNA probes (DIG RNA Labeling Kit (SP6/T7), Roche) and were observed under a microscope (Nikon Eclipse 80i, Tokyo, Japan). Detailed steps were referred to Zhang et al. [31].

### 2.7. Subcellular Localization

The target fragment was amplified using primer P5 containing *Bam*HI and *Xho*I enzyme restriction sites (Table S1). The amplified fragment was connected to pBWA(V)BS-GFP vector using an In-Fusion cloning kit (Vazyme, Nanjing, China). After successful cloning, a strain containing pBWA(V) BS-*BrCAP*-GFP was injected into 4-week-old tobacco leaves. There were three tobacco plants for each treatment, and three to five leaves were treated for each plant. Then, a 24 h dark culture and 24 h light culture was conducted in an MLR-350H incubator (SANYO, Osaka, Japan). The transfected plants were observed using a laser confocal microscope (TCS SP8, Leica, Wetzlar, Germany).

### 2.8. Identification of Disease Resistance of A. thaliana Mutant cap to P. brassicae

The seeds of *A. thaliana* wild type (WT) and mutant *CAP* were sterilized and spread evenly in Petri dishes containing moist filter paper. The seeds were cultured at 25 °C for 2–3 d in the dark until germination. The germinated seeds were cultured in an incubator (60% humidity, 16 h light/8 h dark) for about 3–4 d until the root hair had grown. The seedlings were inoculated with a $1 \times 10^7$/mL *P. brassicae* suspension being dropped on the base of plant roots. WT was served as the control and *CAP* as the treatment. Three plants were randomly selected from WT and *CAP*, respectively, every 24 h after inoculation until the root hairs were found under a microscope (Nikon Eclipse 80i, Tokyo, Japan) to be infected, and the infection rate was investigated.

## 3. Results

### 3.1. Obtainment and Prediction of BrCAP Gene

In our previous study, the results of two-dimensional (2-DE) electrophoresis showed that the expression of G15 protein in the roots of Chinese cabbage inoculated with *P. brassicae* was significantly higher than that in the control roots. Mass spectrometry analysis showed that Bra011464 was highly matched to protein G15 [29]. A BLASTN search found that Bra011464 has a 94% percent identity with *Arabidopsis thaliana CAP*. Thus, this gene was named *BrCAP* (Table 1). The full-length 525 bp coding sequence (CDS) of Bra011464 was cloned from the cDNA of roots from Chinese cabbage, which includes the start codon ATG and stop codon TGA. Sequence analysis found that it is completely consistent with the reference sequence (NC_024795.2) in the *Brassica* database (Figure 1a). RT-qPCR showed that the expression of *BrCAP* in the roots of Chinese cabbage inoculated with *P. brassicae* was 12 times higher than that in the control root (Figure 1b, Table 2). The results showed that the expression of *BrCAP* was significantly changed when Chinese cabbage was infected by *P. brassicae*. The TMHMM was used to analyze the amino acid sequence of *BrCAP*, and the results showed that it had an obvious transmembrane domain (Figure 1c), indicating it might be a membrane protein. A CD Search in NCBI prediction showed that *BrCAP* contained a conserved domain, which was named SCP_PR-1_LIKE (Figure 1d). Therefore, *BrCAP* may play a role in the resistance against clubroot disease, which provides impetus for our subsequent research.

**Table 1.** NCBI Blast Result of Bra011464 cDNA.

| Comparison Result | Percent Identity |
|---|---|
| *Brassica rapa* (XM_009121709.2) | 100% |
| *Brassica napus* (XM_013810782.3) | 99% |
| *Brassica oleracea* var (XM_013730593.1) | 98% |
| *Arabidopsis thaliana* CAP (NM_119530.3) | 94% |

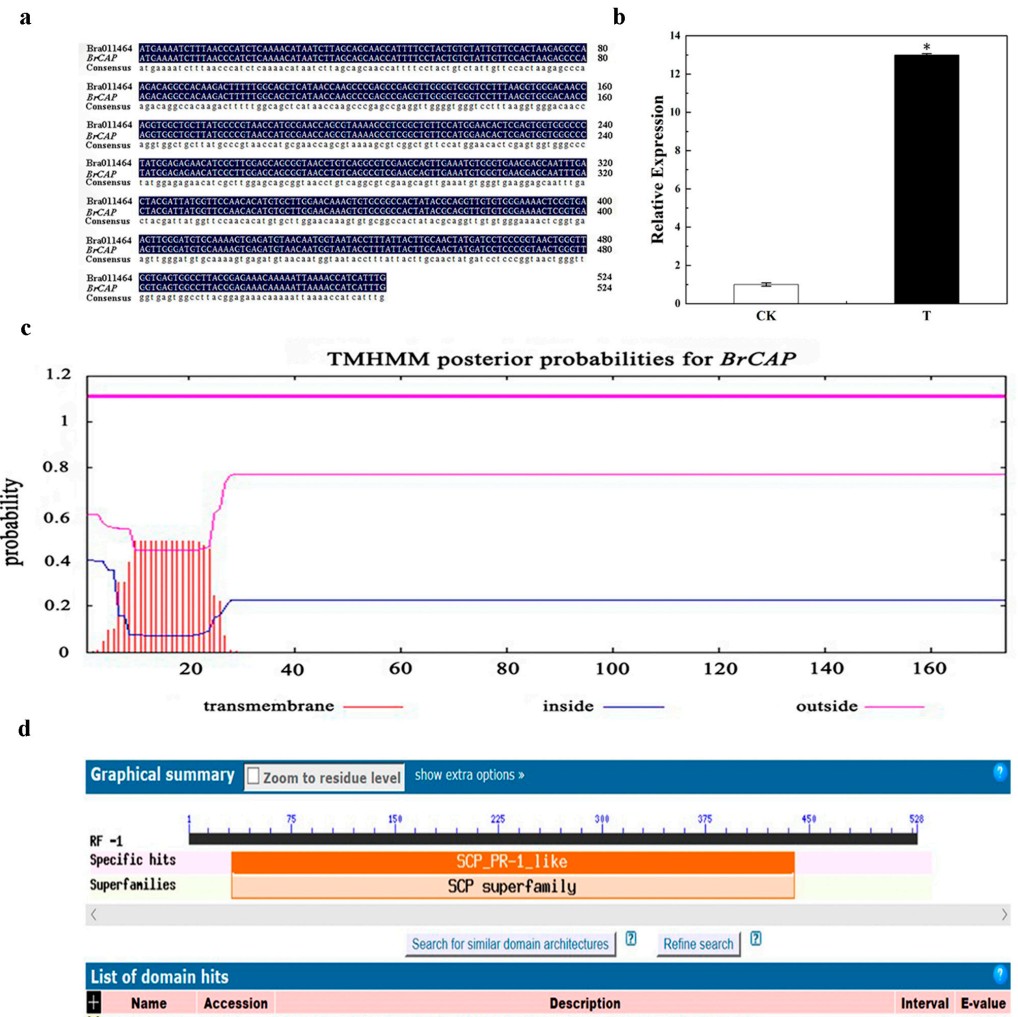

**Figure 1.** Cloning and bioinformatics analysis of *BrCAP*. (**a**). Comparison of full-length CDS clone sequences of *BrCAP*. (**b**). Relative expression of *BrCAP* in the roots inoculated and un-inoculated with *P. brassicae* (Note: CK: Control; T: Treatment (inoculated with *P. brassicae*). Data represent the mean ± standard deviation (*n* = 3). * indicates significant differences at $p \leq 0.05$, according to a Student's *t*-test. (**c**). Transmembrane domain of BrCAP. (**d**). Prediction of conservative domain in BrCAP.

**Table 2.** Expression level of *BrCAP* in *P. brassicae*-inoculated and un-inoculated plants.

| | Expression Level of *BrCAP* | *p* |
|---|---|---|
| CK | 1 ± 0.13 | 0.000 |
| T | 11.73 ± 0.18 * | |

Note: * indicates that the significance was $p = 0.000$, which was at $p \leq 0.05$, according to a Student's *t*-test.

### 3.2. Expression Mode Analysis of BrCAP

The expression of *BrCAP* in roots, stems, and leaves of Chinese cabbage 'SN742' after inoculation and un-inoculation with *P. brassicae* was analyzed by RT-qPCR. The results showed that the expression level of *BrCAP* in the roots inoculated with *P. brassicae* was 10 times higher than that in un-inoculated roots. However, the expression of *BrCAP* in stem and leaf were extremely low (Figure 2a, Table 3). This indicated that *BrCAP* was a root expression gene. Similarly, the expression level of *BrCAP* gene was detected on the 14th and 40th days in roots inoculated with *P. brassicae* by RT-qPCR. The results showed that the expression of *BrCAP* was more than eight times higher in the inoculated roots than that in un-inoculated roots on the 14th day after inoculation. Additionally, its expression was increased 20 times more in the *P. brassicae*-inoculated roots than that in un-inoculated roots on the 40th day (Figure 2b, Table 4). In conclusion, with the accumulation of time, the expression of *BrCAP* in the roots after inoculation with *P. brassicae* was gradually up-regulated and significantly higher than that in the un-inoculated control. The results of RT-PCR were further verified by in situ hybridization, which showed that all the tissue slices of root hybridized with sense probes have no hybridization signal. However, when hybridizing root tissue sections with antisense probes, blue hybridization signals were present on both the 14th and 40th days in the inoculated treatment group, and the signal was stronger on the 40th day. The above results further indicated that *BrCAP* might be related to *P. brassicae* infection (Figure 2c). The results of subcellular localization showed that pBWA(V)BS-*BrCAP*-GFP could produce the green fluorescence signal and co-localize with the red auto-fluorescence signal on chloroplasts of leaf epidermal cells (Figure 2d).

**Table 3.** Statistical analysis of the expression level of *BrCAP* among roots, stems, and leaves.

| Treatment | Expression Level of *BrCAP* | | | Statistical Data | |
|---|---|---|---|---|---|
| | Root | Stem | Leaf | F | p |
| CK | $1 \pm 0.13$ [b] | $0.01 \pm 0.0006$ [c] | $0.01 \pm 0.0013$ [c] | 262.917 | 0.000 |
| T | $11.73 \pm 0.18$ [a] | $0.01 \pm 0.0003$ [c] | $0.01 \pm 0$ [c] | | |

Note: [a–c] indicates that the significance was $p = 0.000$, which was at $p \leq 0.05$, according to Duncan's multiple range test.

**Table 4.** Statistical data of the expression level of *BrCAP* between inoculated treatment with *P. brassicae* and un-inoculated control at different infection periods.

| Period after Inoculation | Expression Level of *BrCAP* | | Statistical Data | |
|---|---|---|---|---|
| | CK | T | F | p |
| 14 days | $1 \pm 0$ | $9.2 \pm 0.07$ | 2.629 | 0.00 |
| 40 days | $1 \pm 0.06$ * | $20 \pm 1.33$ * | 0.130 | 0.00 |

Note: * indicates that the significance was $p = 0.000$, which was at $p \leq 0.05$, according to a Student's *t*-test.

### 3.3. Resistance Identification of Arabidopsis Thaliana Mutant CAP

Phenotypic observation showed that the wild type (WT) grows better than the mutant *CAP* during both growth and bolting stages (Figure 3a). Microscopic observation of root hairs of WT and *CAP* at the 24 h, 48 h, and 72 h after inoculation with *P. brassicae* showed that the spores of *P. brassicae* first infected the fibrous roots of *CAP* at the 48 h, while the fibrous roots of WT were first infected at 72 h. Compared with WT plants, the pathogenic process of *P. brassicae* was accelerated in *CAP*. Thus, *CAP* may play a key role in the progress of plants' resistance against the infection of *P. brassicae* (Figure 3b).

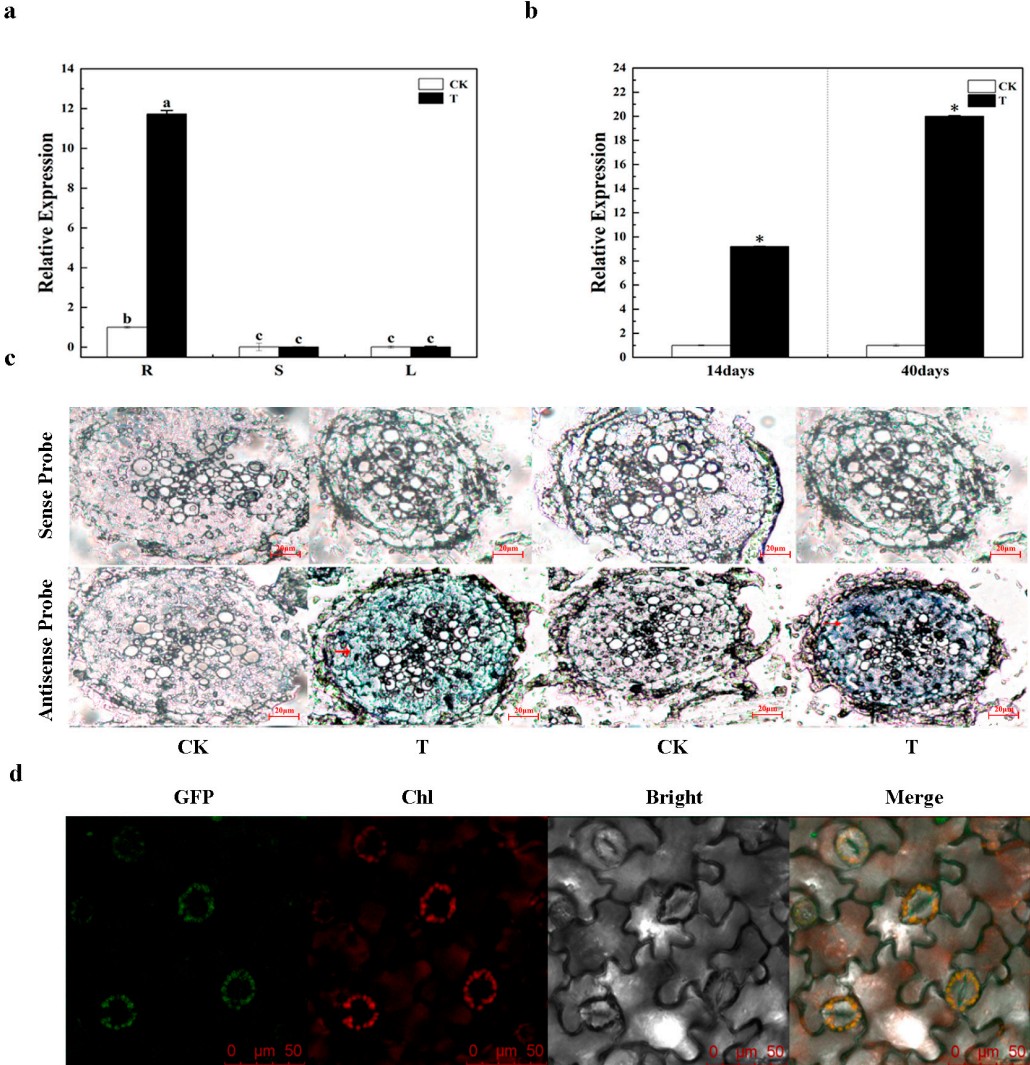

**Figure 2.** Expression mode analysis of *BrCAP*. (**a**). Relative expression of *BrCAP* in different tissues of Chinese cabbage. (R: Root; S: Stem; L: Leaf; CK: control; T: Treatment (the roots inoculated with *P. brassicae*). The data represent the mean ± standard deviation (*n* = 3). a–c means in the same row with different letters differ significantly, $p \leq 0.05$, according to Duncan's multiple range test. (**b**). Relative expression of *BrCAP* on different days after inoculation with *P. brassicae*. The data represent the mean ± standard deviation (*n* = 3). * indicates the significance difference at $p \leq 0.05$, according to Student's *t*-test. (**c**). In situ hybridization of roots on the 40th day after inoculation with *P. brassicae*. The data represent the mean ± standard deviation (*n* = 3). Red arrows show stained tissue. (**d**). Subcellular localization of *BrCAP*. (Green fluorescence represents GFP signal. Red fluorescence represents Chl autofluorescence signal. Bright stands for control and Merge stands for Fusion signal).

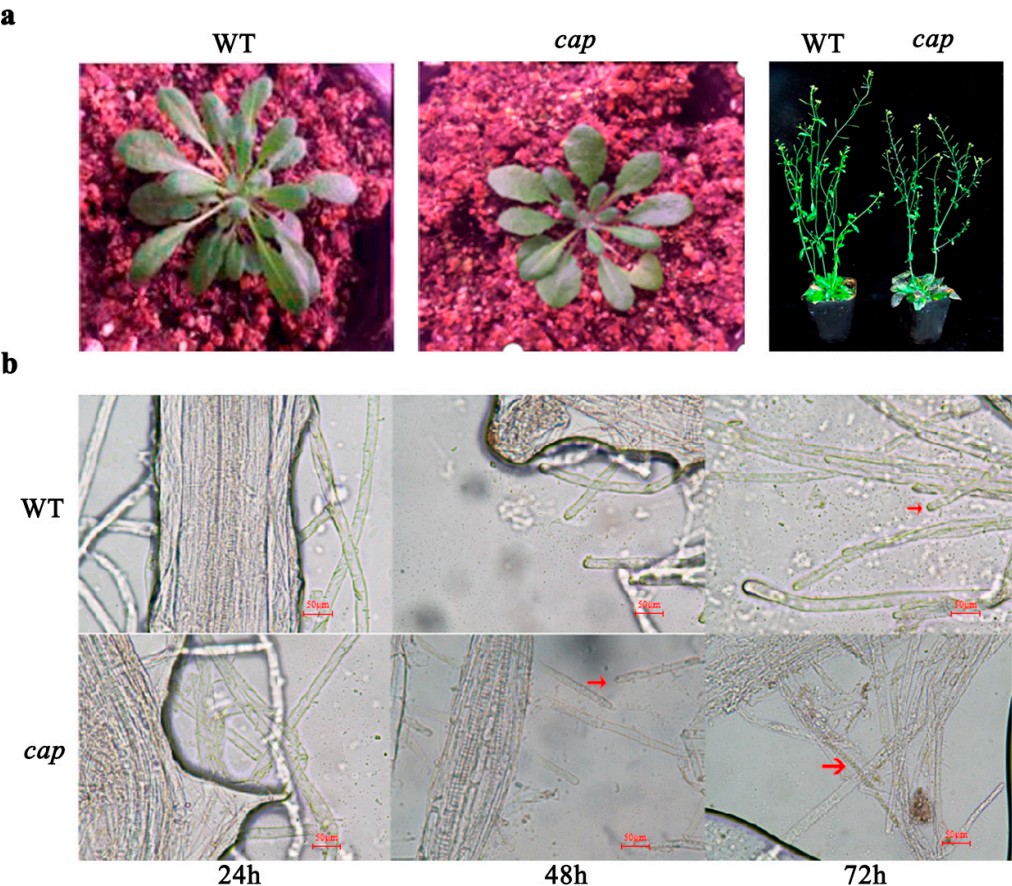

**Figure 3.** Phenotypic comparison and resistance identification for mutant *CAP* and WT. (**a**). Phenotypic comparison. (**b**). Infection rate of the spores of *P. brassicae* for the roots of WT and *CAP* at 24 h, 48 h, and 72 h. The red arrow indicates the site of spore infection.

## 4. Discussion

### 4.1. The Expression of BrCAP in Chinese Cabbage Is Related to the Infection of P. brassicae

Plants exposed to biotic and abiotic stresses can produce a series of responses, such as the expression changes of defense-related genes. The expression level of PR-1 protein was significantly increased after abiotic stress [32]. Most of the *GmPR-1* genes of soybean were up-regulated in response to abiotic stress such as salt and drought [33]. Drought stress caused the up-regulation of all *SlPR-1* genes of up to 50 times, indicating that *SlPR-1* could respond to drought stress [34]. PR-1 protein also responses to biological stress: the expression level of *CsPR-1* of a tea plant was significantly up-regulated under tea blister-blight stress [35]. *ZmPR-1* of maize was significantly up-regulated after infection with *Setosphearia turcica* [36]. Thus, *PR-1* may play a role in the resistance of abiotic stress and biological stress. In our previous study, 16 differential expressed proteins (DEPs) in clubroot-diseased and control roots of Chinese cabbage were screened and identified by two-dimensional electrophoresis and matrix-assisted laser desorption/ionization-time of fight/mass spectrometry. One of them was Bra011464 (PREDICTED: pathogenesis-related protein 1, *Brassica rapa*) [30]. In this study, Bra011464 was found to have a 94% percent identity with *Arabidopsis thaliana* CAP, named *BrCAP* (Table 1). RT-qPCR analysis showed that the expression level of *BrCAP* in roots was significantly higher than that in stems and leaves (Figure 2a), which indicated that *BrCAP* was a root expression gene. The results of RT-qPCR and in situ hybridization showed that the expression level of *BrCAP* was significantly higher in inoculated roots than that in the control (Figure 2a). Its expression difference was more significant on the 40th day after inoculation (Figure 2b,c). Therefore, the expression of *BrCAP* in Chinese cabbage is related to the infection of *P. brassicae*.

### 4.2. BrCAP was Localized on Chloroplasts

In cells, the subcellular localization of a protein is closely related to its molecular functions. Determining the subcellular localization of a protein provides valuable clues with which to trace its molecular function [37]. Currently, the commonly methods of subcellular localization mainly include agrobacterium-mediated transient transformation [38], particle bombardment [39], and protoplast transformation [40]. PR-1 was a member of CAP superfamily proteins [26]. The study showed PR1-positive compartments were found to be highly mobile and of variable shape [41,42]. When the infusion protein pCamA-*TaLr35PR1*-GFP was introduced into onion epidermal cells by particle bombardment method, the green fluorescence was mainly concentrated outside the cells, indicating that *TaLr35PR1* gene encodes an extracellular protein. Additionally, *TaLr35PR1* mainly plays an extracellular role [43]. Tamara found the AtPR1 protein is localized on a vesicle-like intracellular compartment [27]. However, there are no reports about the location of the CAP family. In this study, we found that *BrCAP* was localized on chloroplasts by agrobacterium-mediated method (Figure 2d), which can be used to trace its molecular function.

### 4.3. BrCAP May Play a Key Role in the Resistance of Chinese Cabbage against Clubroot Disease

CAP (Cysteine-rich secreting protein, Antigen 5, and disease-related Protein 1) is a key gene in response to attack and infection [18], and PR-1 was a member of CAP superfamily proteins [26]. PR-1 is a disease-course related protein that has been studied in tomatoes, bananas, soybeans, and tea trees [34,35]. PR-1 is an SA receptor that is involved in the SA-dependent defense response [44]. Studies have shown that transgenic tobacco plants with the *PR-1A* gene introduced had a high level of compositional expression and strong resistance against tobacco black shank disease and downy mildew [45]. In addition, *PR-1a* (from tobacco) inhibited the growth of the oomycete pathogen *Phytophthora brassicae*, which proved that PR-1 proteins have anti-oomycete properties [46]. *cap1-1* mutants exhibit elevated levels of reactive oxygen species (ROS) under $NH^{4+}$ stress, and increased the expression of respiratory burst oxidase homologous genes and decreased the expression of catalase gene compared with the wild type. In addition, *cap1-1* mutants produce smaller branches and smaller epidermal cells in response to $NH^{4+}$ emphasis [47–49]. However, no relevant studies have shown that *BrCAP* can play a role in resistance against clubroot disease. In this study, we compared the infection rate of wild type (WT) and mutant *CAP*, and it was found that *CAP* was infected with *P. brassicae* during the 48 h and WT was infected during the 72 h, indicating that *CAP* was more susceptible to infection than WT and the absence of *BrCAP* could positively regulate the infection of *P. brassicae* (Figure 3b). Our previous study showed that the interactions between Chinese cabbage and *P. brassicae* stimulate the SA signaling pathway and the content of SA and the expression of genes in the SA signaling pathway were altered in the clubroot diseased roots [29]. These results suggested that *BrCAP* may play a role in the resistance response of Chinese cabbage against clubroot disease by inducing the SA signal pathway.

### 5. Conclusions

In a previous study, we found the up-regulated gene *BrCAP*. In this study, RT-qPCR and in situ hybridization showed that the expression level of *BrCAP* was higher in the roots than in stems and leaves of Chinese cabbage. The roots after inoculation with *P. brassicae* were further found to be more significant than the control. These results indicated that *BrCAP* was a root expression gene, and its expression in roots was up-regulated after inoculation with *P. brassicae*. Subcellular localization showed *BrCAP* located on chloroplasts of leaf epidermal cells. In addition, the *A. thaliana* deletion mutant *cap* can be infected more easily by *P. brassicae* than by *Arabidopsis* wild type (WT), which indicated that the deletion of the *CAP* gene could promote the infection of *P. brassicae*. Therefore, it was suggested that *BrCAP* plays a key role in the resistance response of Chinese cabbage against clubroot disease.

**Supplementary Materials:** The following supporting information can be downloaded at: https://www.mdpi.com/article/10.3390/horticulturae9040517/s1, Table S1: Primers for all manuscript.

**Author Contributions:** J.Z. (Jiawei Zou): conceptualization, methodology, writing—original draft, formal analysis, investigation, writing—review and editing. S.G.: methodology, formal analysis, data curation, investigation. B.Z.: formal analysis, data curation. W.G.: resources; formal analysis. J.Z. (Jing Zhang): visualization, software; R.J.: funding acquisition, conceptualization, formal analysis, supervision, resources, writing—review and editing. All authors have read and agreed to the published version of the manuscript.

**Funding:** This research was funded by [The National Natural Science Foundation of China] grant number [31972412, 32272717].

**Data Availability Statement:** No new data were created.

**Acknowledgments:** The authors would like to thank Xiangqun Shen for providing technical guidance.

**Conflicts of Interest:** We declare that we have no known competing financial interest or personal relationships that could have appeared to influence the work reported in this paper.

## Abbreviations

| | |
|---|---|
| *P. brassicae* | *Plasmodiophora brassicae* Woronin |
| RT-qPCR | Real-time quantitative polymerase chain reaction |
| 2-DE | two-dimensional electrophoresis |
| CAP | Cysteine-rich secreting protein, Antigen 5, and disease-related Protein 1 |
| *A. thaliana* | *Arabidopsis thaliana* |
| WT | wild type |

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
