# Peer review of "Chinese Cabbage BrCAP Has Potential Resistance against Plasmodiophora brassicae"

_horticulturae, doi:10.3390/horticulturae9040517_

Round 1
Reviewer 1 Report
In the current manuscript, the authors isolated and expressed the BrCAP protein and studied its resistance capacity with Plasmodiophora brassicae. The authors claimed that the BrCAP plays a role in the resistance response of Chinese cabbage against clubroot disease. This conclusion is only based on the expression analysis. However, it is not clear if only BrCAP is expressed higher or if there could be other proteins expressed in a significant amount. I have other points that could help to improve the manuscript.
1. In the expression analysis, how the authors measured the expression levels of BrCAP and other proteins? This needs to be quantitatively done instead of qualitatively.
2. In the abstract, the authors mentioned “In this study, a differentially expressed protein, BrCAP, was found between the roots of Chinese cabbage inoculated and uninocualted with P. brassicae through two-dimensional electrophoresis (2-DE) and Mass spectrometry analysis.” I did not find any results of 2D and mass spectrometry.
3. qPCR and cloning analysis need to be clearer and should have more details.
4. In situ hybridization section, the authors performed five replicates for each different treatment. What are those? In the same section, the authors indicated “Specific steps were refered to Zhang et al. (2021)” The authors may want to state some specifics here and cite the reference.
5. Figures 2 and 3 need to be discussed in detail as these could potentially explain the importance of BrCAP in clubroot disease and help to draw conclusions systematically.
Minor points:
1. Elaborate on the conclusions.
2. Improve the quality of figures.
3. What is figure 2D show? Why it is important?
Author Response
The manuscript has been rechecked and the necessary changes have been made in accordance with your suggestions. The responses to all comments have been prepared and given below. All revisions are highlighted in yellow for easy identification in file of manuscript with mark.
In the current manuscript, the authors isolated and expressed the BrCAP protein and studied its resistance capacity with Plasmodiophora brassicae. The authors claimed that the BrCAP plays a role in the resistance response of Chinese cabbage against clubroot disease. This conclusion is only based on the expression analysis. However, it is not clear if only BrCAP is expressed higher or if there could be other proteins expressed in a significant amount. I have other points that could help to improve the manuscript.
Point 1: In the expression analysis, how the authors measured the expression levels of BrCAP and other proteins? This needs to be quantitatively done instead of qualitatively.
Response 1:
Thank you very much for your proposal. The expression levels of BrCAP and other proteins were measured by 2D and mass spectrometry, which was showed in our previous study, detailed in the article ‘’The salicylic acid signaling pathway plays an important role in the resistant process of Brassica rapa L. ssp. pekinensisto Plasmodiophora brassicae Woronin” (Ji et al., 2020). The relevant revises have been highlighted in yellow (Lines 10-13).
Point 2: In the abstract, the authors mentioned “In this study, a differentially expressed protein, BrCAP, was found between the roots of Chinese cabbage inoculated and uninocualted with P. brassicae through two-dimensional electrophoresis (2-DE) and Mass spectrometry analysis.” I did not find any results of 2D and mass spectrometry.
Response 2:
I'm sorry for our false statement. These results were included in our previous study, detailed in the article “The salicylic acid signaling pathway plays an important role in the resistant process of Brassica rapa L. ssp. pekinensisto Plasmodiophora brassicae Woronin” (Ji et al., 2020). The relevant revises have been highlighted in yellow (Lines 10-14).
Point 3: qPCR and cloning analysis need to be clearer and should have more details.
Response 3:
Thank you very much for your suggestion. We accept the expert's advice and have made the results of qPCR and cloning analysis more details. The revised portion was highlighted in yellow in the manuscript (Lines 262-273 and 302-320).
Point 4: In situ hybridization section, the authors performed five replicates for each different treatment. What are those? In the same section, the authors indicated “Specific steps were referred to Zhang et al. (2021)” The authors may want to state some specifics here and cite the reference.
Response 4:
‘The five replicates’ means that five slices are used for each root tissues, and each slice has three to five tissues. The revised portion was highlighted in yellow in the manuscript (Lines 220-222 and 229).
Point 5: Figures 2 and 3 need to be discussed in detail as these could potentially explain the importance of BrCAP in clubroot disease and help to draw conclusions systematically.
Response 5:
Thank you very much for your suggestion. Figures 2 and 3 have been discussed in detail (Lines 302-341, 366-379,421-428, 444-446 and 482-487).
Minor points:
- Elaborate on the conclusions.
Response: the conclusions have been revised. (Line 494-507)
- Improve the quality of figures.
Response: the quality of figures has been improved to 600DPI.
- What is figure 2D show? Why it is important?
Response: The answers are in line 339-341 and 444-446.
All Minor points have been revised

Reviewer 2 Report
The manuscript addresses a very interesting and highly valuable topic in terms of understanding plant-pathogen interaction. The study is well justified and approached, it offers very relevant data and information due to its interest for the scientific community and even the producers. Also, the manuscript is consistent with the journal. However, I find in the current version several aspects that can be improved, these were indicated in the manuscript file. But I also noticed three aspects that in my opinion are critical:
1) The authors should mention their experimental design with greater clarity and level of detail, indicating their treatments, replicates and measurement factors.
2) The authors must include a section on statistics, and say how their data explored and how they analyzed it, mentioning the model and statistical tests they used, including significance levels.
3) Include, as part of your results, the values of test statisticians (eg. F-ratios) for each factor, either through a table or in the text, before pointing a posteriori tests such as Duncan's or t-student. Likewise, it is recommended to improve the presentation of your photographs by indicating what they show in terms of structures.
In Discussion: It seems to me that there should be a hypothesis in the introduction that would lead the discussion, the discussion seems brief, I think that the authors can exploit their data more and offer explanations for their results before offering suggestions.

Author Response
The manuscript has been rechecked and the necessary changes have been made in accordance with your suggestions. The responses to all comments have been prepared and given below. All revisions are highlighted in yellow for easy identification in file of manuscript with mark.
The manuscript addresses a very interesting and highly valuable topic in terms of understanding plant-pathogen interaction. The study is well justified and approached, it offers very relevant data and information due to its interest for the scientific community and even the producers. Also, the manuscript is consistent with the journal. However, I find in the current version several aspects that can be improved, these were indicated in the manuscript file. But I also noticed three aspects that in my opinion are critical:
Point 1: The authors should mention their experimental design with greater clarity and level of detail, indicating their treatments, replicates and measurement factors.
Response 1: Thank you very much for your suggestion. The experimental design has been mentioned with greater clarity and level of detail. Please find the revise from the manuscript with mark (Lines:128-131; Lines :159-169; Lines:200-209; Lines:220-221; Lines:235-237; Line:246-249).
Point 2: The authors must include a section on statistics, and say how their data explored and how they analyzed it, mentioning the model and statistical tests they used, including significance levels.
Response 2: Thank you very much for your suggestion. The section on statistics had been added (Lines:166-169).
Point 3: Include, as part of your results, the values of test statisticians (eg. F-ratios) for each factor, either through a table or in the text, before pointing a posteriori tests such as Duncan's or t-student. Likewise, it is recommended to improve the presentation of your photographs by indicating what they show in terms of structures.
Response 3: The values of test statisticians (eg. F-ratios) for each factor has been added (Table 2-4; Line 294, 360and362; Lines 307-308? Line 316-317), and the presentation of your photographs had been improved (Line355 and 386)
In Discussion: It seems to me that there should be a hypothesis in the introduction that would lead the discussion, the discussion seems brief, I think that the authors can exploit their data more and offer explanations for their results before offering suggestions.
Response: Discussion had been improved (Lines 406-446 and463-492).

Reviewer 3 Report
Dear authors!
My recommendations:
1) Induction by hormone treatment is not always stressful event (page 2).
2) Have you used the terms verbalized, verbalization correctly (page 3)? I have only met them in texts about learning process.
3) Write the sentences "Water permeably every 3-4 days", "Primer P1 (Table S1) and ... Kit." (page 3), "Using TMHMM... Structure domain" , "After the incubation of 24 h in the dark ... incubator", "Seeds accelerate ... environment" (page 4), "To examine the expression ... and inoculated plants" (page 6) more understandable. Is permeably could be changed to applied?
4) Add some information in the Methods. How long are the germinating seeds were cultured in the incubator? What was the age of inoculated plants (page 4)?
5) I have seen from Fig. 1b that the difference of the expression level is nearly 12/1=12 (page 5), from Fig 2B - the difference is nearly 10-20 (page 6), not 5 .
6) What is the B stress mean (page 8)?
7) You should not repeat some sentences in the Introduction (page 2) and the Discussion (page 8).
My recommendations concerning the manuscript design:
1) Shorten the term Plasmodiophora brassicae (Abstract, page 2) for the first time.
2) Write with a small letter: mass, in situ (Abstract), injury, salt, induction (page 2), bioinformatics (page 3), structure (page 4).
3) Write in italics: PROAtCAPE1 (page 2), Nicotians benthamiana, Botrytis cinerea, PR-1A (page 8).
4) Remove the dot (Title, at the end of 4.1. and 4.2. sentences), the words P. brassicae (page 3), Acknowledgments (page 8).
5) Write with large letters TAIR, Petri (page 3).
6) Use the uniform font size (Abstract, page 3, page 4)
7) Decipher for the first time: CD (page 5).
8) References should be described according instructions for authors.
Author Response
The manuscript has been rechecked and the necessary changes have been made in accordance with your suggestions. The responses to all comments have been prepared and given below. All revisions are highlighted in yellow for easy identification in file of manuscript with mark.
Point 1: Induction by hormone treatment is not always stressful event (page 2)
Response 1:
Thank you very much for your proposal. We accept the expert's advice and have removed the relevant part (Lines 75-76).
Point 2: Have you used the terms verbalized, verbalization correctly (page 3)? I have only met them in texts about learning process
Response 2:
Sorry, only verbalization is correct, we have modified the relevant part (Line 145).
Point 3: Write the sentences "Water permeably every 3-4 days", "Primer P1 (Table S1) and ... Kit." (page 3), "Using TMHMM... Structure domain", "After the incubation of 24 h in the dark ... incubator", "Seeds accelerate ... environment" (page 4), "To examine the expression ... and inoculated plants" (page 6) more understandable. Is permeably could be changed to applied?
Response 3:
Thank you very much for your suggestion. We have modified the relevant part and the revised portion was highlighted in yellow in the manuscript. The sentence "Water permeably every 3-4 days" has been deleted. This sentence "Primer P1 (Table S1) and ... Kit." has changed (Lines160-164). "Using TMHMM... Structure domain" has changed (Lines 210-213). "After the incubation of 24 h in the dark ... incubator" has changed (Lines 128-131). This sentence "To examine the expression ... and inoculated plants" writing has changed (Lines 302-306). The relevant parts have been highlighted in yellow.
Point 4: Add some information in the Methods. How long are the germinating seeds were cultured in the incubator? What was the age of inoculated plants (page 4)?
Response 4:
Thank you very much for your proposal. The seeds were cultured at 25℃ for 2-3 d in the dark until germination. The germinated seeds were cultured in an incubator (60% humidity,16h-light/8h-dark) for about 3-4d until the root hair was grow, the seedlings were inoculated (Line 245-249).
Point 5: I have seen from Fig. 1b that the difference of the expression level is nearly 12/1=12 (page 5), from Fig 2B - the difference is nearly 10-20 (page 6), not 5.
Response 5:
Thank you very much for your proposal. We made a mistake in stating and we have modified the relevant part (Lines 269, 305).
Point 6: What is the B stress mean (page 8)?
Response 6:
It’s Boron, I’m sorry to use abbreviations. But we have delated the part in revised manuscript.
Point 7: You should not repeat some sentences in the Introduction (page 2) and the Discussion (page 8).
Response 7:
Thank you very much for your proposal. The repeat sentences we have delated. And we rewrite the discussion section.
The manuscript design:
Point 1: Shorten the term Plasmodiophora brassicae (Abstract, page 2) for the first time.
Response 1:
We have changed in Line 8.
Point 2: Write with a small letter: mass, in situ (Abstract), injury, salt, induction (page 2), bioinformatics (page 3), structure (page 4).
Response 2:
We have changed in Line13, 19, 75,76,197,213.
Point 3: Write in italics: PROAtCAPE1 (page 2), Nicotians benthamiana, Botrytis cinerea, PR-1A (page 8).
Response 3:
We have changed PROAtCAPE1 in Line 70, Nicotians benthamiana in Line 100. Botrytis cinerea has been deleted. PR-1A has changed in Line 473.
Point 4: Remove the dot (Title, at the end of 4.1. and 4.2. sentences), the words P. brassicae (page 3), Acknowledgments (page 8).
Response 4:
These part have been revised.
Point 5: Write with large letters TAIR, Petri (page 3).
Response 5:
We have changed accordingly in Line 98,128.
Point 6: Use the uniform font size (Abstract, page 3, page 4)
Response 6:
These parts have been revised.
Point 7: Decipher for the first time: CD (page 5).
Response 7:
We have changed in Line 262.
Point 8: References should be described according instructions for authors.
Response 8:
These parts have been revised.

Round 2
Reviewer 1 Report
The authors revised the manuscript and addressed all the issues satisfactorily. The manuscript is much improved and can be accepted.